# The effect of COVID-19 vaccination in Italy and perspectives for living with the virus

Valentina Marziano [1,2,7], Giorgio Guzzetta [1,2,7], Alessia Mammone[3], Flavia Riccardo [4], Piero Poletti [1,2], Filippo Trentini [1,2,5], Mattia Manica [1,2], Andrea Siddu[3], Antonino Bella[4], Paola Stefanelli [4], Patrizio Pezzotti [4], Marco Ajelli [6,8], Silvio Brusaferro[4,8], Giovanni Rezza[3,8] & Stefano Merler [1,2,8✉]

COVID-19 vaccination is allowing a progressive release of restrictions worldwide. Using a mathematical model, we assess the impact of vaccination in Italy since December 27, 2020 and evaluate prospects for societal reopening after emergence of the Delta variant. We estimate that by June 30, 2021, COVID-19 vaccination allowed the resumption of about half of pre-pandemic social contacts. In absence of vaccination, the same number of cases is obtained by resuming only about one third of pre-pandemic contacts, with about 12,100 (95% CI: 6,600-21,000) extra deaths (+27%; 95% CI: 15–47%). Vaccination offset the effect of the Delta variant in summer 2021. The future epidemic trend is surrounded by substantial uncertainty. Should a pediatric vaccine (for ages 5 and older) be licensed and a coverage >90% be achieved in all age classes, a return to pre-pandemic society could be envisioned. Increasing vaccination coverage will allow further reopening even in absence of a pediatric vaccine.

[1] Center for Health Emergencies, Bruno Kessler Foundation, Trento, Italy. [2] Epilab-JRU, FEM-FBK Joint Research Unit, Trento, Italy. [3] Health Prevention Directorate, Ministry of Health, Rome, Italy. [4] Istituto Superiore di Sanità, Rome, Italy. [5] Dondena Centre for Research on Social Dynamics and Public Policy, Bocconi University, Milan, Italy. [6] Laboratory for Computational Epidemiology and Public Health, Indiana University School of Public Health, Bloomington, United States. [7]These authors contributed equally: Valentina Marziano, Giorgio Guzzetta. [8]These authors jointly supervised: Marco Ajelli, Silvio Brusaferro, Giovanni Rezza, Stefano Merler. ✉email: merler@fbk.eu

Since December 2020, vaccination against COVID-19 is being rolled out in all countries of the world, in a race to put an end to the devastating effects of the pandemic in terms of lives lost[1], hospital congestion[2], economic disruption[3], and mental health[4]. While African countries are painfully struggling to have access to vaccines and to distribute them (only 4.5% of the population in Africa is fully vaccinated, as of October 1, 2021[5]), most high-income countries had a fast deployment, with over half of their citizens being fully immunized by July 2021[5]. Thanks to the high efficacy and effectiveness of the licensed vaccines against SARS-CoV-2 infection, severe disease, and death[6–10], and to the prioritization of the highest risk categories, these countries were able to limit the damages caused by the emergence of the hypertransmissible Delta variant[11–13]. For example, in the European Union, despite ample relaxations of physical distancing restrictions conceded by governments over the summer of 2021, the peak mortality never exceeded 1.5 deaths per million (as of October 31, 2021) since Delta become dominant in July, as compared to values over three times higher from November 2020 through April 2021[5]. Similarly, the incidence of confirmed cases remained within about 150 cases per million, a value that is lower than those systematically observed between mid-October 2020 and mid-May 2021[5]. With the ongoing progress of immunization campaigns, there is a need to quantitatively assess their impact on health and social activities, as well as to evaluate potential future epidemiological scenarios. In particular, as the emergence of the Delta variant has severely dwindled chances to eliminate SARS-CoV-2[14] in countries that have not managed to maintain a zero-COVID approach[15], there is a need to identify strategic objectives towards "living with COVID-19"[16] at least in the medium term.

In this study, we use a mathematical model of SARS-CoV-2 transmission, informed by detailed real-world data, to retrospectively evaluate the effect of COVID-19 vaccination in Italy during the first half of 2021, and to prospectively assess potential future scenarios associated to different coverage levels.

## Results

We adapted an age-structured, compartmental model of SARS-CoV-2 transmission in Italy[17,18] that estimates the level of social activity needed to match the net reproduction number, as computed from official epidemic curves recorded in the national integrated surveillance system[10,19]. The level of social activity is expressed in terms of the proportion of social contacts measured before the pandemic[20]. The model keeps into account the dynamics of age-specific population immunity due to both infection[17], the progression of the vaccination campaign[21], and the waning of immunity. We assume that protection from both natural and vaccine-induced immune response wanes exponentially with a baseline average duration of 2 years[22,23]. We assume that successfully vaccinated individuals are not fully immune ("leaky vaccine") with different efficacy values for preventing infection and lethal disease. We tuned the model with data from the initial phase of the vaccination campaign (December 27, 2020–June 30, 2021), when the SARS-CoV-2 Alpha variant was dominant in the country[24], and we project model results for the future by considering the progression of the vaccination campaign and the dominance of the Delta variant as of October 2021[25,26]. Further details on the model are provided in Section Methods.

**Retrospective analysis**. The model reproduces the observed number of COVID-19 cases and deaths in vaccinated (partially or fully) and unvaccinated individuals over the first half of 2021 (Fig. 1A–C). Considering the population immunity acquired from both vaccination and infection, a significant fraction of the Italian

population (36.2%, 95% CI: 35.9–36.7%) was estimated to be fully susceptible to SARS-CoV-2 as of June 30, 2021, with high heterogeneity by age (Fig. 1D). This population immunity profile would have been insufficient to avoid potential successive outbreaks if caution was not applied when lifting physical distancing restrictions; for example, a complete resumption of pre-pandemic social activity would result in an effective reproduction number of 1.9 (95% CI: 1.8–2.1) on June 30, 2021, even in absence of the more transmissible Delta variant.

To evaluate the impact of the COVID-19 vaccination program in Italy over the first half of 2021, we simulated a scenario where we assume that the actual epidemic trajectory would be maintained, in absence of vaccination, by an appropriate reduction in social activity over time, due to both governmental measures and individual behavioral choices. Under these hypotheses, a decrease of about one fourth—from 48% (95% CI: 44–51%) to 35% (95% CI: 33–37%)—of the average proportion of active social contacts at the end of June 2021 would have been needed (Fig. 2A). Furthermore, we estimate that about 12,100 additional deaths (95% CI: 6600–21,000, corresponding to an increase of 27%, 95% CI: 15–47%), would have occurred in the population even under the same cumulative number of cases (Fig. 2B), mostly because of a larger proportion of infections among the high-risk segments of the population. Finally, we estimate that the potential for successive waves would be much larger due to the lower population immunity under this scenario, with an estimated effective reproduction number of 2.6 (95% CI: 2.4–2.8) (Fig. 2C).

If the Alpha variant had remained dominant until September 2021, we estimated that the progress of the vaccination campaign in July and August 2021 (Fig. 3A) would have resulted in a decline of the reproduction number from 0.92 (95% CI: 0.88–0.95) on June 30 to 0.61 (95% CI: 0.54–0.71) on September 7, 2021 (Fig. 3B). However, the Delta variant had rapidly replaced Alpha in July 2021[25]. Considering a 50% increase in transmissibility of the Delta variant[11–13], the estimated reproduction number on September 7, 2021, is 0.91 (95% CI: 0.81–1.06), close to the observed value of 0.83 (95% CI: 0.82–0.84)[27]. Thus, the increased transmissibility of the new variant and the progress of the vaccination campaign in the summer of 2021 have essentially leveled out, resulting in similar values of the reproduction number at the end of June and the beginning of September, 2021.

**Future vaccination scenarios**. We projected the potential impact of a further future progression of the vaccination campaign. To this aim, we evaluated scenarios in which all age groups will reach a given coverage Ω; age groups, which were already above that coverage on September 7, 2021, will remain unaffected (see the schematic example on Fig. 4A). We then projected the reproduction number for different values of Ω and different proportions of pre-pandemic contacts that are resumed (Fig. 4C). A complete return to the pre-pandemic lifestyle would still result in reproduction numbers significantly higher than the epidemic threshold of 1 and is therefore unlikely to achieve, even with an almost complete coverage of the population aged 12+ years. This is due to the high transmissibility of the Delta variant and the imperfect protection against infection granted by vaccination. However, expanding the coverage would allow a significant resumption of social activity while maintaining the reproduction number under the epidemic threshold (Fig. 4B), from a 56% (95% CI: 45–62%) of pre-pandemic contacts estimated for a coverage >60% in all age classes (close to the uptake level already achieved on September 7, 2021) to a projected 76% (95% CI: 48–100%) for a 100% coverage of the eligible age groups (12 years or older). If a pediatric vaccine (for children aged 5 years and older) will be

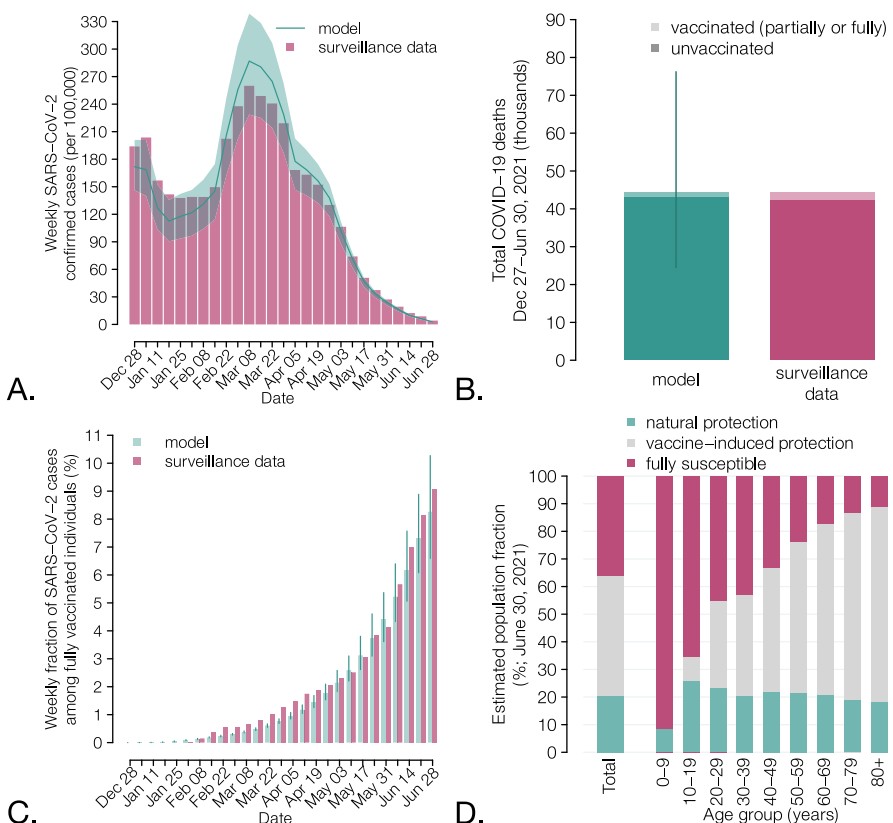

**Fig. 1 Characteristics of the COVID-19 epidemic in Italy during the first half of 2021. A** Weekly incidence per 100,000 population of SARS-CoV-2 confirmed cases (the x axis reports the starting day of the considered week); bars: data from the Italian Integrated Surveillance System[50]; line and shaded area: mean and 95% CI of the model estimates; $n = 300$ stochastic model realizations. **B** Total number of COVID-19 deaths over the study period (in thousands) among vaccinated (partially or fully) and unvaccinated individuals. Green: mean (bar) and 95% CI (vertical lines) of the model estimates ($n = 300$ stochastic model realizations); red: data from the Italian Integrated Surveillance System[45]. **C** Weekly percentage of confirmed SARS-CoV-2 cases occurring in fully vaccinated individuals over the total. The fraction of cases in completely vaccinated individuals increases over time because of the progressive increase in the vaccinated population. Green: mean (bar) and 95% CI (vertical lines) of the model estimates ($n = 300$ stochastic model realizations); red: data from the Italian Integrated Surveillance System[45]. **D** Mean estimates of the immunity profile of the Italian population, overall and by age groups, on June 30, 2021 ($n = 300$ stochastic model realizations). Individuals who have been infected after being vaccinated or who have been vaccinated despite still having a protection from infection are counted under the natural protection bar; individuals who have never been infected or who have lost their natural protection and were vaccinated (partially or fully) are included under the vaccine-induced protection bar; individuals who were never vaccinated nor infected, or who were infected but lost their natural protection, or who were vaccinated but lost their vaccine-induced protection are included under the fully susceptible bar.

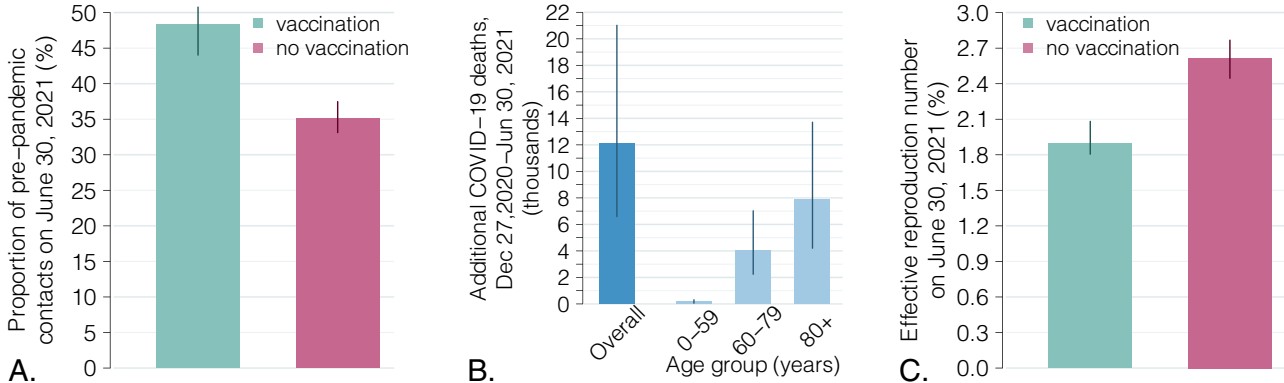

**Fig. 2 Impact of the vaccination program during the first half of 2021. A** Estimated active social contacts on June 30, 2021, as a proportion of pre-pandemic contacts, with and without a vaccination program, under the constraint that the two scenarios reproduce the same observed epidemic trajectory. Bars: mean estimates; vertical lines: 95% CI; $n = 300$ stochastic model realizations. **B** Number of additional COVID-19 deaths between December 27, 2020, and June 30, 2021, total and by age group, under a no-vaccination scenario. Bars: mean estimates; vertical lines: 95% CI; $n = 300$ stochastic model realizations. **C** Effective reproduction number (i.e., under complete resumption of pre-pandemic contacts) on June 30, 2021, with and without vaccination. Bars: mean estimates; vertical lines: 95% CI; $n = 300$ stochastic model realizations.

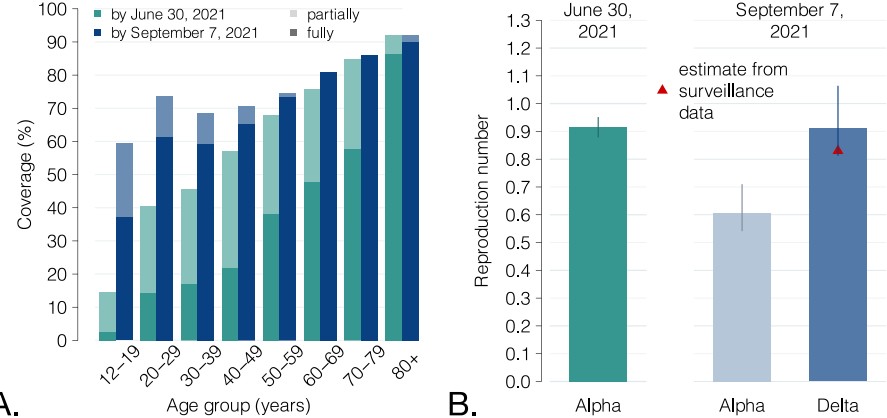

**Fig. 3 Vaccination coverage by June 30 and September 7, 2021, and effect of the replacement of the Alpha variant by the Delta variant. A** Comparison between the fraction of the Italian population that was partially and fully vaccinated by June 30, 2021, and by September 7, 2021, by age group. **B** Green: net reproduction number on June 30, 2021, when the Alpha variant was still largely dominant. Mean and 95% CI as reported in[45]. Light blue: estimated value of the reproduction number, given the progression of the vaccination program until September 7 and under the assumption that the Alpha variant remained dominant; bars: mean estimates; vertical lines: 95% CI; n = 300 stochastic model realizations. Dark blue: the same effect under the assumption of a 50% increase in transmissibility to reproduce the replacement of the Alpha variant with the Delta, occurred during the summer of 2021[11–13,25]; bars: mean estimates; vertical lines: 95% CI; n = 300 stochastic model realizations. Red triangle: value of the reproduction number as estimated from surveillance data[27]; for this estimate, the 95% CI is not visible at the scale of the plot.

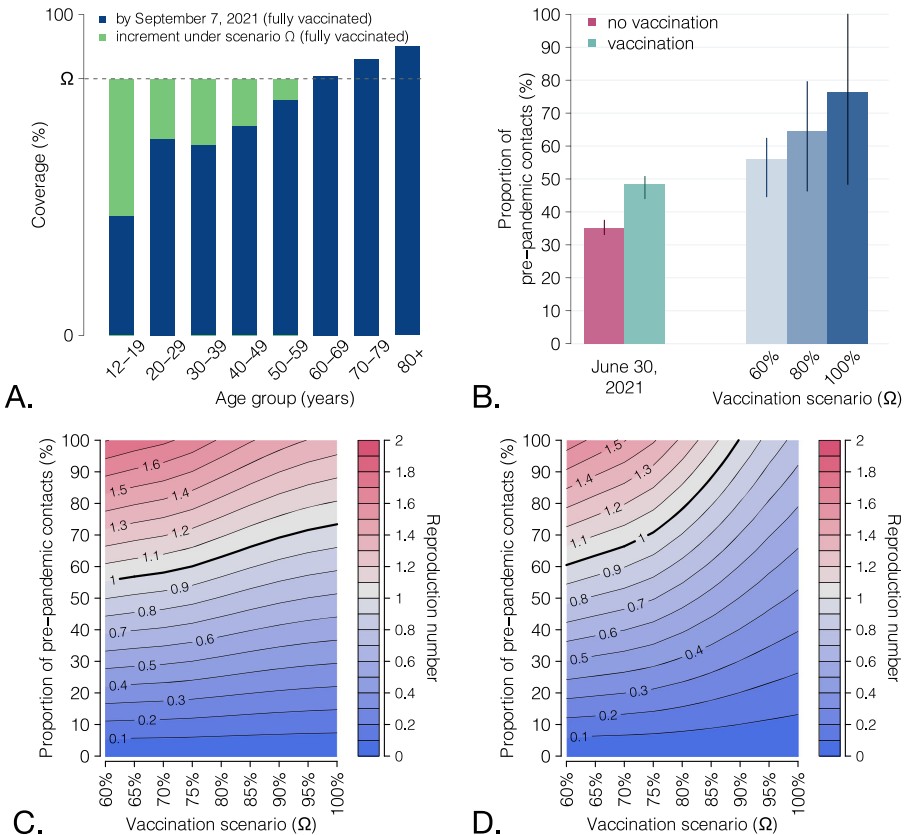

**Fig. 4 Scenarios for the expansion of vaccination coverage. A** Schematic of simulated scenarios. All age classes with coverage below a given value Ω are assumed to progress to Ω; all age classes above Ω will remain at the coverage level achieved on Sep 7. **B** Proportion of pre-pandemic contacts corresponding to a reproduction number of 1 for three selected vaccination scenarios; levels estimated to be active on June 30, 2021 (with and without vaccination) are reported for comparison. Bars: mean estimates; vertical lines: 95% CI; n = 300 stochastic model realizations. **C** Heatmap of the mean estimated reproduction number for different vaccination scenarios (x axis) and different levels of social activity (y axis); n = 300 stochastic model realizations. Contour lines discriminate different values of the reproduction number. The thicker contour line represents the epidemic threshold of 1. **D** As **C**, but assuming that coverage Ω is achieved also in pediatric age groups (5–11 years).

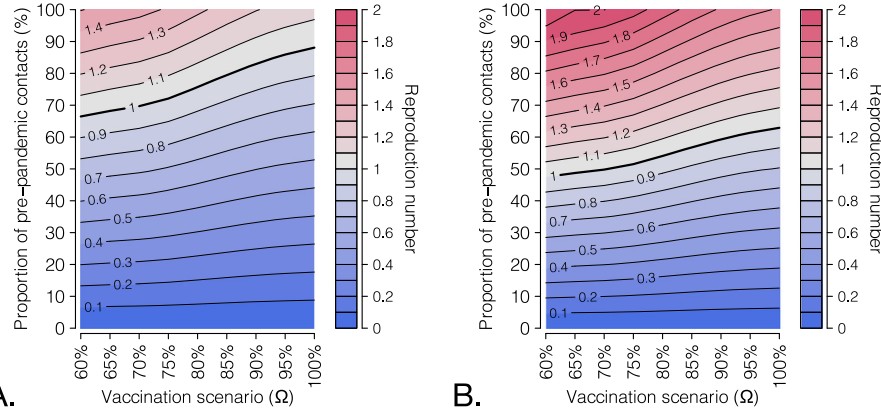

**Fig. 5 Sensitivity analysis with respect to the transmissibility increase of the Delta with respect to the Alpha variant.** Heatmap of the mean projected reproduction number for different vaccination scenarios (x axis) and different levels of social activity (y axis), n = 300 stochastic model realizations, under a transmissibility increased by **A**) 25% and **B**) 75%, compared to the Alpha variant.

licensed and widely distributed, we projected that herd immunity may be reached even for a complete return to pre-pandemic social behavior with a coverage of at least 90% in all age classes (Fig. 4D). Such herd immunity, however, would be only temporary, due to the waning vaccine protection over time.

We analyze the sensitivity of the estimated prospective reproduction numbers against different values for the increase in transmissibility of Delta compared to Alpha (and in absence of a pediatric vaccine). We show that for a transmissibility increase of 25%, the proportion of pre-pandemic contacts that could be resumed without causing an epidemic would increase to 65–85%, depending on the coverage scenario (Fig. 5A). If Delta is 75% more transmissible than Alpha, the corresponding range would be limited to 45–60% (Fig. 5B).

## Discussion

In this work, we quantified the retrospective and prospective impact of the COVID-19 vaccination campaign in Italy, which kicked off on December 27, 2020. First, we show that in the first half of 2021, a similar epidemic trajectory in absence of the vaccine would have resulted in a 27% (95% CI 15–47%) excess of COVID-19 deaths compared to the ones observed in the same period. This would have also required a reduction of social activity by one quarter (from 48 to 35% of pre-pandemic contacts at the end of June). In addition, a much higher risk for further waves of infection would be maintained, with an average effective reproduction number on June 30, 2021, of 2.6, instead of the 1.9 estimated in the presence of vaccination. Second, our results suggest that the replacement of the Alpha variant (and of other lineages) with the more transmissible Delta variant during the month of July was offset by the progression of the vaccination campaign in the months of July and August, resulting in a value of the reproduction number in early September that was similar to the one estimated at the end of June 2021. Summer vaccinations, however, did not reduce the transmissibility alone, but also the risk of severe disease and death in the population, given the high effectiveness of vaccines against these endpoints; thus, despite the Delta variant, the epidemiological outlook at the start of September 2021 was likely better than that at the end of June 2021. Finally, our results show that the future epidemic trend is surrounded by substantial uncertainty. We estimate that expanding the vaccine coverage could allow a further increase of social activity while maintaining the reproduction number below the epidemic threshold. However, the high transmissibility of Delta and the imperfect vaccine protection against infection could

not be sufficient for a complete return of society to the pre-pandemic life without the risk of occurrence of further pandemic waves. Our results are in line with previously published modeling studies investigating the interplay between vaccination and relaxation of control measures[28,29,30].

Should a pediatric vaccine (for ages 5 and older) be licensed and a coverage >90% be achieved in all age classes, and assuming a 50% higher transmissibility of the Delta variant, a complete return to pre-pandemic society could still be envisioned. For these estimates, we assume that between September 7, 2021, and the time the coverage for that vaccination scenario has been reached, the alteration of the population immunity profile due to the opposite forces of waning immunity and of the continued circulation of SARS-CoV-2 will be negligible, compared to that caused by the progression of vaccination. This assumption may be broken if large waves of COVID-19 occur before reaching the considered coverage or if enough time elapses (several months) for a substantial waning of immunity. However, the administration of booster doses that is taking off in the fall of 2021 in countries with a high population coverage, including Italy[31], will likely reduce the risks related to waning immunity.

One limitation of this study is that we implicitly assumed that vaccinated and unvaccinated individuals have the same probability of contacting each other. However, it is known that vaccine hesitancy clusters spatially and demographically[32], increasing the probability of local outbreaks in undervaccinated pockets even when the average reproduction number is below the epidemic threshold. To explore this effect, data on the clustering of COVID-19 vaccine hesitancy are warranted.

We did not consider the effect on our results of other features of the Delta variant, such as its potential ability to escape natural immunity[33–35], which is still partially undefined. In addition, even the increased transmissibility of Delta is subject to several unknowns; its value was estimated in situations where physical distancing restrictions were broadly active and thus a large proportion of interactions were with close contacts[12,13]. It is possible that, as interventions relax and social contacts increases, the estimated transmission advantage of Delta over Alpha (about 50%) will be different. In a sensitivity analysis, we showed that the actual value of this parameter critically affects epidemiological prospects.

The dynamics of loss of protection over time for different population demographics (age, comorbidities) and clinical endpoints (infection, death, transmissibility of breakthrough infections) will likely affect future COVID-19 trajectories and must be better elucidated with long-term follow-up studies. Based on

preliminary studies[22,23], we assumed an average duration of two years for the protection conferred by both infection and vaccination, and equal for all individuals. In sensitivity analyses, we show that different durations of the natural immunity may affect our estimates of the effective reproduction number at the end of June 2021 and therefore impact prospective scenarios for the next year (see Supplementary Figs. 10 and 13). Similarly, if breakthrough infections were as transmissible as infections in unvaccinated individual, this would increase the estimated reproduction numbers and reduce the levels of societal reopening that would be affordable (see Supplementary Figs. 10–11).

This work highlights the multiple epidemiological and social benefits allowed by the vaccination efforts in terms of averted deaths, reopening of social activity, and reduced risks of further epidemic waves. In addition, our study shows the potential for further resuming social activities granted by the expansion of vaccination coverage in the perspective of "living with the virus". In particular, the availability of pediatric vaccines, which, as of October 2021, are under scientific investigation and regulatory scrutiny[36], could greatly contribute to societal reopening should the coverage be sufficiently high. However, the acceptability of a pediatric vaccine may be limited by the perceived small risk of COVID-19 disease in children, especially if adverse vaccine events will be recorded even with very low rates[37].

The scenario of complete resumption of pre-pandemic social life would entail removing all the persisting factors that today still reduce the number of an individual's contacts compared to the pre-COVID-19 era. These include residual governmental limitations (e.g., capacities in stadiums and discotheques, number of people who can be seated together at restaurants indoors, etc.); organizational measures reducing crowding (e.g., the capacity of workplace spaces and the use of work from home, distancing of desks in schools, mandatory booking for recreational and cultural activities, regulations for weddings and other large events); social distancing etiquette; and individual choices to reduce one's own risks of infection. In addition, several preventive measures further reduce the contacts that are important for transmission (those considered in the model) without significantly affecting social interactions, e.g., mandates for EU digital COVID-19 certificate[38] (currently required in Italy for accessing workplaces, schools, and indoor recreational facilities), ventilation policies and air filtering systems on public transport, plexiglass separators between restaurant tables or at counters of commercial and public offices, face masks, and testing, tracing and isolation protocols. Although quantifying the impact of each of these measures and norms is extremely hard, it is likely that many of them will linger for a long time without a significant negative influence on either the economy or the social life of individuals. Therefore, a complete resumption of pre-pandemic contacts in the sense considered by the model may not necessarily be a key objective. Depending on the measures that will be maintained, on the acquired coverage, and assuming a 50% higher transmissibility of the Delta variant, we estimate the SARS-CoV-2 reproduction number to take values between 0.7 (if contacts will not increase and coverage will be close to 100%) and 1.8 (if social activity will be fully resumed and norms will be abandoned without increases in vaccine coverage). If the reproduction number is slightly above 1, hospitalizations could remain limited and the impact on hospitals manageable provided that frail individuals are sufficiently protected by the vaccine.

Finally, we stress that our prospective results need to be revised in case of the future emergence of new hypertransmissible variants. Such a possibility could jeopardize the gains afforded by vaccination programs, forcing new setbacks in the recovery of social contacts, and exacerbating the burden of a potential further epidemic resurgence.

## Methods

We developed an age-structured stochastic model of SARS-CoV-2 transmission and vaccination, based on a susceptible-infectious-removed-susceptible (SIRS) scheme (Supplementary Fig. 1)[17,18]. The population is stratified by age (17 5-year age groups from 0 to 84 years plus one age group for individuals aged 85 years or older) and presence/absence of comorbidities (Supplementary Fig. 3). Mixing patterns are encoded by an age-specific social contact matrix estimated prior to the COVID-19 pandemic[20]. Susceptibility to SARS-CoV-2 infection is assumed to be age-dependent (lower in children under 15 years of age and higher for the elderly above 65 years, compared to individuals aged 15–65)[39]. Infectiousness was assumed to be homogeneous by age groups and symptomatic status[39,40]. We consider a basic reproduction number $R_0$ for historical lineages of 3.0[19,41,42]. The model was used to simulate the vaccination campaign and the evolution of COVID-19 epidemiology in Italy between December 27, 2020 (start of vaccination) and June 30, 2021. Throughout this period, the dominant variant was Alpha[24]; therefore, in our retrospective investigation we considered an increase in transmissibility by 50% compared to historical lineages[24,43,44].

The rollout of the vaccination campaign is modeled using detailed data on the daily age-specific number of doses administered over the considered period (Supplementary Fig. 6)[21]. Individuals are considered eligible for vaccination, independently of a previous diagnosis of SARS-CoV-2 infection. To account for preferential administration of different types of vaccines by age group, we estimated the age-specific vaccine efficacy against infection by weighting the efficacy of a specific vaccine type (mRNA vs. viral vectors) by the number of vaccines of that type administered to each age group (Supplementary Fig. 7)[21], considering a vaccine efficacy against infection of 89% after two doses of mRNA vaccine, and of 62% after two doses of viral vector vaccine[8,9]. The efficacy against death was set to 80.6% in partially and 96.4% in fully vaccinated individuals[45]. Breakthrough infections (i.e., infections in vaccinated individuals) were assumed to be half as infectious as those in unvaccinated individuals;[46,47] we additionally considered a sensitivity analysis where the infectiousness is the same. Immune protection is assumed to wane after an exponentially distributed time (average 2 years[22,23] in the baseline for both the natural and vaccine-induced protection; alternative values are considered as sensitivity analyses). After waning of protection, individuals are considered fully susceptible.

To reproduce the epidemic curve over the study period, we adjusted a scaling factor representing the proportion of pre-pandemic contacts that were active on a given day, in such a way that the model's reproduction number (estimated via the Next Generation Matrix approach[48,49]) would match the corresponding estimate from surveillance data (Supplementary Fig. 2)[10]. We compared estimates obtained with the actual vaccination rollout against those that would be required in a hypothetical scenario without vaccination to obtain the same epidemic curve. We evaluated the prospective impact of the vaccination campaign by considering the replacement of the Alpha with the Delta variant (occurred in July 2021)[25], which is assumed to be 50% more transmissible[11–13] (transmissibility increases of 25 and 75% are considered in sensitivity analyses). We also considered vaccination scenarios where the age-specific vaccination coverage achieved by September 7, 2021, is incremented for all age classes that were below a given target coverage $\Omega$ at that date and left unchanged for age classes above. For each scenario, we estimate reproduction numbers under different degrees of resumption of pre-pandemic contacts (from 0 to 100%). Full model details are reported in Supplementary Information.

**Reporting summary.** Further information on research design is available in the Nature Research Reporting Summary linked to this article.

## Data availability

Time-series of cases and deaths reported to the National Integrated Surveillance System are available at https://www.epicentro.iss.it/coronavirus/open-data/covid_19-iss.xlsx[50]. Data on the daily age-specific number of doses administered in Italy by vaccine type are available at https://github.com/italia/covid19-opendata-vaccini[21]. The input data used for model simulations are available on Zenodo at https://doi.org/10.5281/zenodo.5703240[51].

## Code availability

The code used in this study is available on Zenodo at https://doi.org/10.5281/zenodo.5703240[51].

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

## Acknowledgements

V.M., G.G., F.R., P.Po., F.T., M.M., P.Pe., and S.M. acknowledge funding from EU grant 874850 MOOD (catalogued as MOOD 020). V.M., G.G., P.Po., and S.M. acknowledge funding from the Fondazione Valorizzazione Ricerca Trentina (VRT), project COVID-VAX. The contents of this publication are the sole responsibility of the authors and don't necessarily reflect the views of the funders.

## Author contributions

V.M., M.A., S.B., G.R., and S.M. designed research; V.M., G.G., P.Po., F.T., and M.M. performed research; V.M., G.G., A.M., F.R., A.S., A.B., P.S., and P.Pe. analyzed data; G.G. drafted the first version of the manuscript; and V.M., G.G., P.Po., P.Pe., M.A., and S.M. wrote the paper. All authors contributed to data interpretation, critical revision of the manuscript, and approved the final version of the manuscript.

## Competing interests

M.A. has received research funding from Seqirus. The funding is not related to COVID-19. All other authors declare no competing interest.
