## [Peer Review File · Nature Communications]

Reviewers' Comments:

Reviewer #1:

Remarks to the Author:

This paper describes a simulation study exploring the conditions in which physical distancing restrictions (PDRs) could be relaxed under various vaccination scenarios while keeping case counts relatively constant.

Overall, I thought the study was well written and well executed. Figures were complex but (for the most part) easy to understand for the amount of information being conveyed. My main concern was whether the dimensions of uncertainty included here are the ones that will make the most difference. For example, I feel uneasy about the claims of "zero COVID" mere months away. Of course, models are simplifications, but we more or less know this won't happen, so the fact that the model makes this prediction is more of a model limitation than a finding!

As a second example, while the authors have laudably included uncertainty, the uncertainty bounds are quite narrow in many cases (see Fig. 1, for example). This is partly due to the assumption that "cases are maintained at 50 per 100,000". My feeling is that actual countries, Italy included, came nowhere near close to controlling it this precisely. As this is one of the most central assumptions in the paper, it should be more fully justified (it's not even mentioned in the appendix why this level was chosen, it seems). I see a sensitivity analysis has been run in Fig. S19, but given the centrality of this assumption, perhaps slightly more attention could be paid to this.

Minor comments:

- Line 35: "around month 14" feels a little jarring since surely this is extremely dependent on the assumptions surrounding waning immunity.
- Line 49: These countries have also (almost?) universally employed highly effective contact tracing programs, not just lockdowns and border controls.
- Line 53: "an alternance between periods" -- perhaps "alternating periods"?
- Line 75: I'm not sure I understand why the graphs start at -40% rather than 0%? From this description it sounds like they should start (more or less) at 0%. Maybe rather than this nonintuitive relative measure, 0% could represent no contact and 100% could represent pre-COVID levels of contact? That might be easier to interpret than ranges of ~140% with an unclear baseline.
- Line 124: Could you clarify the waning assumptions a bit more here? E.g. is it full protection for 6-24 months, then zero? Or exponential?
- Fig. 3: It may be worthwhile revisiting the caption text (e.g., "high risk" rather than "fragile").
- Fig. 4: Why does duration of vaccine protection seemingly have no effect? I am also surprised the transmissibility scenarios don't have more impact. That zero COVID can be achieved so quickly, even in e.g. the complete reopening scenario, is also surprising, but I suppose this doesn't consider heterogeneities or immune-escaping variants. It might be worth making the assumption that natural immunity wanes at the same rate as vaccine-derived immunity, rather than that it's permanent (which it doesn't seem to be).
- Line 207: Perhaps better to say "Italy" rather than "countries" since my understanding is that other contexts were not parameterized.
- Supplement, line 166: It's unclear why the main text refers to Italy and the supplement to Lazio.

Reviewer #2:

Remarks to the Author:

In this paper, authors explore the conditions for a return to normal under COVID-19 mitigation measures and vaccinations. I find that some of the results are interesting even though authors need to do substantially more to ensure that their findings can be translated more easily into the real-world context, that they used the latest data on vaccine efficacy and describe the assumptions they use.

1) Assumptions about the vaccine distribution rate: I think it's currently difficult to translate the results into something Public Health officials can relate to. For example, 41 million doses have already been administered in Italy. Readers will want to know what that corresponds to in terms of potential R in the absence of PDRs, i.e. where is Italy positional in the road to reopening. But they cannot determine that from the paper as it only provide key quantities as a function of the time elapsed since the start of the vaccination. However, this time is calculated under the assumption the daily number of people being vaccinated will remain constant over time. This clearly has not been the case, with an increase in vaccine roll out in many countries similar to Italy. For example, the current vaccination rate in Italy appears to be twice that used in the model. As a result, the timeline provided by the model seems wrong, with the coverage achieved in Italy being larger than that anticipated by the model at a given time. It seems essential authors find a way to help readers understand what it means to be x months after the start of vaccination, for example by showing the associated vaccine coverage. They also need to carefully think about the relevance of a scenario with a constant vaccination rate, when this is clearly not the case. This is also the reason why the abstract which provides conditions for reopening within 9 to 15 months may seem not completely rooted in real-world.

2) Assumptions about vaccine effectiveness: Authors only account for the fact that vaccines reduce the risk of infection. There is now also evidence that vaccines provides a large protection against the risk of severe infection (higher than that obtained against the risk of infection) and that it also reduces infectivity. It seems important that authors account for the last data on vaccine effectiveness in their assessments. They should revise their model assumptions accordingly.

3) Description of assumptions in the main text. The description of the model and its assumption in the main text is too short; and readers need to read the Supplement to understand the work being done. It is important that key information are provided in the main text. For example, the assumptions about vaccine effectiveness, which are central to the work, are only provided in the Supplement. Similarly, which population is targeted for vaccination? Is it really assumed that the vaccination coverage will be the same in older individuals and in young adults/teenagers/children if vaccinated? What are the assumptions about R0 for the historical virus and B.1.1.7? When authors talk about a vaccination coverage of 75% is it across the whole population? >6 years old?

4) Authors quantify the effort required to maintain the number of cases constant with their variable "relaxation of PDRs" that is equal to 0 to do the work for the historical virus and 100% corresponds to complete relaxation. I think that for many readers it will be difficult to relate to this variable. For example, does a relaxation of 0 corresponds to the Italian lockdown of the first wave? Or less stringent measures? I wonder if it wouldn't be easier to follow if authors presented the reduction in transmission rates that is required, 0 would mean no contact at all, 100% would be all contacts restored.

5) Authors should comment in discussion about risks associated with other variants, in particular the delta variant.

6) What are current vaccination coverages by age group in Italy? Isn't the one among older individuals already above 75%? And if so, would it be relevant to increase it?

7) Figure 4 remains difficult to follow.

Other comments:

Abstract "A short-lived vaccine protection will require re-vaccination campaigns in absence of which a new intensification of PDRs is expected around month 14". Presumably, the month at which this occurs will be highly dependent on the duration of vaccine protection and the rhythm at which the population was vaccinated. Authors should either detail their assumptions or make a statement that remains more general.

Figure 1 Average relaxation of PDRs over 2 years is the average over time of the variable represented in Panel 1B, is that correct?

Appendix line 128: "Astra Zeneca vaccine for the population under 55 years" in many countries this vaccine is no longer distributed to those aged <55 years. Revise?

It looks like, in their baseline scenario, authors only consider the impact of the vaccine on the risk

of infection, not severity. There is now good evidence that SARS-CoV-2 vaccines are very effective against severe disease. It is essential that authors account for this characteristic in their main analysis.

Appendix line 192: Vaccination capacity of 4 per 1000, which corresponds to about 250,000 doses per day for a country of the size of Italy. A potential limitation with this approach is that the vaccination rate has increased dramatically in the last few months. While this rate was elevated for many European countries some time ago, current vaccination rates can now be much higher than this.

Line 66: please give R_0 assumed for the historical lineage

Line 84: the number of death after 2 years without vaccination is completely dependent on the daily number of cases assumed by authors right? What was the rationale to select that number?

Figure 3B: I don't understand why, if we completely reopen at the time when the only the most fragile have been vaccinated, we have a huge drop in R , that goes below 1. Is there a problem with the way the Figure was plotted? Or is it really the case that it would be a good idea to relax everything at this stage?

In Figures, authors need to indicate what a relaxation=0 represents.

REVIEWER COMMENTS

Reviewer #1 (Remarks to the Author):

This paper describes a simulation study exploring the conditions in which physical distancing restrictions (PDRs) could be relaxed under various vaccination scenarios while keeping case counts relatively constant.

Overall, I thought the study was well written and well executed. Figures were complex but (for the most part) easy to understand for the amount of information being conveyed.

We thank the reviewer for appreciating our work and for the useful comments provided. Given the evidence available at this time, we deeply restructured the study to provide indications more rooted in the real-world context as requested by both reviewers.

The model is now informed with:

- official estimates of the daily net reproduction number at the national level, derived from surveillance data;
- data on the daily number of doses administered by age-group to realistically simulate vaccination rollout;
- updated estimates of the vaccine efficacy against infection, death, and infectiousness of breakthrough cases.

In addition, we modified the model to take into account additional features of COVID-19 epidemiology and vaccination, namely:

- waning of immunity acquired either by natural infection or vaccination;
- updated estimates on the age-specific vaccine effectiveness against infection;
- increased effectiveness of vaccines against death with respect to infection;
- the reduced infectiousness of breakthrough infections.

The revised manuscript includes:

- 1) a retrospective analysis of the impact of vaccination in Italy between December 27, 2020 (date of launch of the vaccination campaign) and June 30, 2021; here, the definition of a counterfactual to evaluate the impact of vaccination is not simple, since simply removing vaccination and letting the epidemics evolve in absence of interventions would be completely unrealistic. Therefore, we chose as counterfactual a scenario where governmental restrictions and individual behavior would result in the same epidemic trajectory as the one observed in the presence of vaccination. We show that in absence of vaccination, the same epidemic curve could have been allowed only by more significant reductions in social activity (over one quarter less on June 30, 2021), but, at the same time, it would result in over one quarter more deaths, and would expose the country to much higher risks of subsequent large epidemic waves.
- 2) an assessment of the combined effect of the replacement of the Alpha variant by the Delta variant during the month of July and of the progression of the vaccination campaign in July and August, 2021; we show that the two effects substantially compensated, resulting in the same reproduction number at the end of June and in early September;
- 3) prospective scenario simulations in which we assess how different coverage targets may impact the reproduction number of COVID-19 in Italy. In particular, we show that expanding the vaccine coverage will allow a further, but not complete, resumption of social activity while maintaining the reproduction number below the epidemic threshold. If a pediatric vaccine (for ages 5 and older) is

licensed and a coverage >90% is achieved in all age classes, a complete return to pre-pandemic society could be envisioned.

Accordingly, we have changed the title into “The effect of COVID-19 vaccination in Italy and perspectives for ‘living with the virus’ “.

My main concern was whether the dimensions of uncertainty included here are the ones that will make the most difference. For example, I feel uneasy about the claims of "zero COVID" mere months away. Of course, models are simplifications, but we more or less know this won't happen, so the fact that the model makes this prediction is more of a model limitation than a finding!

We agree with the reviewer that the evolution of the epidemiological situation since the time of writing the manuscript (in particular with the global diffusion of the Delta variant) has made the zero-COVID scenario extremely unlikely. Indeed, in our new analyses we show that the effective reproduction number of SARS-CoV-2 (which quantify the transmission potential expected when all social contacts are resumed) can be brought below the epidemic threshold only with extremely high vaccine coverage (>90%) in all age classes including pediatric ones (5+ years old), for which a vaccine is not even licensed yet. Any mentions of “zero-COVID” have now been removed from the manuscript.

As a second example, while the authors have laudably included uncertainty, the uncertainty bounds are quite narrow in many cases (see Fig. 1, for example). This is partly due to the assumption that "cases are maintained at 50 per 100,000". My feeling is that actual countries, Italy included, came nowhere near close to controlling it this precisely. As this is one of the most central assumptions in the paper, it should be more fully justified (it's not even mentioned in the appendix why this level was chosen, it seems). I see a sensitivity analysis has been run in Fig. S19, but given the centrality of this assumption, perhaps slightly more attention could be paid to this.

We thank the reviewer for this useful comment. We acknowledge that this assumption was one of the main limitations of the submitted work, which was of a purely prospective nature. Given the evidence available at this time, we restructured our work to provide indications more rooted in the real-world context. Specifically, we now adjust the number of social contacts over time in such a way that the reproduction number of the model reflects the official estimates derived from surveillance data. In this way, the model captures the weekly incidence of SARS-CoV-2 cases notified to the Italian integrated surveillance system (Fig 1A in the revised manuscript, reported below for the reviewer’s convenience).

Minor comments:

- Line 35: "around month 14" feels a little jarring since surely this is extremely dependent on the assumptions surrounding waning immunity.
- Line 49: These countries have also (almost?) universally employed highly effective contact tracing programs, not just lockdowns and border controls.
- Line 53: "an alternance between periods" -- perhaps "alternating periods"?

These sentences have been removed in the revised manuscript.

- Line 75: I'm not sure I understand why the graphs start at -40% rather than 0%? From this description it sounds like they should start (more or less) at 0%. Maybe rather than this nonintuitive relative measure, 0% could represent no contact and 100% could represent pre-COVID levels of contact? That might be easier to interpret than ranges of ~140% with an unclear baseline.

We thank the reviewer for the useful suggestion, which we have adopted for the revised manuscript. We now refer to the proportion of pre-pandemic social contacts that are active (range 0-100%). The interpretation of this quantity is also more inclusive, as it accounts for changes in contacts due not only to adjustments of governmental restrictive measures (defined as physical distancing restrictions, PDRs, in the initial submission), but also to individual choices linked to risk perception.

- Line 124: Could you clarify the waning assumptions a bit more here? E.g. is it full protection for 6-24 months, then zero? Or exponential?

We apologize for being unclear on this point. We assume an exponential distribution for the duration of protection. Before waning, vaccine efficacy reduces the probability of infection and death (with different efficacy estimates for the two endpoints) and the infectiousness for breakthrough infections. After waning, individuals are considered fully susceptible. We now specify this explicitly in the Methods.

- Fig. 3: It may be worthwhile revisiting the caption text (e.g., "high risk" rather than "fragile").
- Fig. 4: Why does duration of vaccine protection seemingly have no effect? I am also surprised the transmissibility scenarios don't have more impact. That zero COVID can be achieved so quickly, even in e.g. the complete reopening scenario, is also surprising, but I suppose this doesn't consider heterogeneities or immune-escaping variants.

These figures have now been removed.

It might be worth making the assumption that natural immunity wanes at the same rate as vaccine-derived immunity, rather than that it's permanent (which it doesn't seem to be).

We thank the reviewer for this suggestion, which we have integrated in the updated version of the model. Natural immunity is assumed to confer full protection from infection until it wanes with an exponential distribution. Due to limited quantitative evidence on the duration of natural and vaccine-induced protection, in the baseline analysis we assume an average of 2 years, based on preliminary studies [Hall et al., Lancet, 2021; Andrews et al., medRxiv, 2021]. Alternative durations of 1 year and 10 years are explored as sensitivity analyses.

- Line 207: Perhaps better to say "Italy" rather than "countries" since my understanding is that other contexts were not parameterized.

We limited our comments to Italy. We thank the reviewer for the careful reading.

- Supplement, line 166: It's unclear why the main text refers to Italy and the supplement to Lazio.

We apologize for being unclear. Our simulations started when the vaccination campaign was launched in Italy (end of December 2020). Estimates for the fraction of immune population in Italy at that time are not available; we had adapted a previously published model [Marziano et al, PNAS, 2021] to obtain estimates of the fraction of immune population at the end of 2020 in three different Italian regions, that are representative of different levels of SARS-CoV-2 circulation during 2020:

- Lombardy, the first and hardest hit region in 2020;
- Lazio, characterized by an intermediate circulation of SARS-CoV-2 in 2020
- Campania, characterized by low circulation of SARS-CoV-2 in 2020.

The obtained fraction of immune individuals at the end of 2020 ranged from 9% in Campania to 24% in Lombardy, while the central value obtained for Lazio was around 16%. The estimates obtained for Lazio are used to initialize the fraction of immune population at the national level in the baseline analysis; those for the two other regions were used in sensitivity analyses (lower immunity: Campania; higher immunity: Lombardy) which demonstrated the robustness of results with respect to this initialization. From the epidemic curves of 2020 and under the assumption of an exponentially distributed duration of protection, we estimated that about 8.9% of infected individuals who acquired the infection during 2020 may have lost their immunity by the end of the year (see Appendix), and we reduced the initial immune population accordingly (e.g., for Lazio we considered a fraction of immune population on December 27, 2020, of $16\% \cdot (1 - 0.089) = 14.6\%$).

Reviewer #2 (Remarks to the Author):

In this paper, authors explore the conditions for a return to normal under COVID-19 mitigation measures and vaccinations. I find that some of the results are interesting even though authors need to do substantially more to ensure that their findings can be translated more easily into the real-world context, that they used the latest data on vaccine efficacy and describe the assumptions they use.

We thank the reviewer for appreciating our work and for the useful comments provided. Given the evidence available at this time, we deeply restructured the study to provide indications more rooted in the real-world context as requested by both reviewers.

The model is now informed with:

- official estimates of the daily net reproduction number at the national level, derived from surveillance data;
- data on the daily number of doses administered by age-group to realistically simulate vaccination rollout;
- updated estimates of the vaccine efficacy against infection, death, and infectiousness of breakthrough cases.

In addition, we modified the model to take into account additional features of COVID-19 epidemiology and vaccination, namely:

- waning of immunity acquired either by natural infection or vaccination;
- updated estimates on the age-specific vaccine effectiveness against infection;
- increased effectiveness of vaccines against death with respect to infection;
- the reduced infectiousness of breakthrough infections.

The revised manuscript includes:

- 1) a retrospective analysis of the impact of vaccination in Italy between December 27, 2020 (date of launch of the vaccination campaign) and June 30, 2021; here, the definition of a counterfactual to evaluate the impact of vaccination is not simple, since simply removing vaccination and letting the epidemics evolve in absence of interventions would be completely unrealistic. Therefore, we chose as counterfactual a scenario where governmental restrictions and individual behavior would result in the same epidemic trajectory as the one observed in the presence of vaccination. We show that in absence of vaccination, the same epidemic curve could have been allowed only by more significant reductions in social activity (over one quarter less on June 30, 2021), but, at the same time, it would result in over one quarter more deaths, and would expose the country to much higher risks of subsequent large epidemic waves.
- 2) an assessment of the combined effect of the replacement of the Alpha variant by the Delta variant during the month of July and of the progression of the vaccination campaign in July and August, 2021; we show that the two effects substantially compensated, resulting in the same reproduction number at the end of June and in early September;
- 3) prospective scenario simulations in which we assess how different coverage targets may impact the reproduction number of COVID-19 in Italy. In particular, we show that expanding the vaccine coverage will allow a further, but not complete, resumption of social activity while maintaining the reproduction number below the epidemic threshold. If a pediatric vaccine (for ages 5 and older) is licensed and a coverage >90% is achieved in all age classes, a complete return to pre-pandemic society could be envisioned.

Accordingly, we have changed the title into “The effect of COVID-19 vaccination in Italy and perspectives for ‘living with the virus’ “.

1) Assumptions about the vaccine distribution rate: I think it’s currently difficult to translate the results into something Public Health officials can relate to. For example, 41 million doses have already been administered in Italy. Readers will want to know what that corresponds to in terms of potential R in the absence of PDRs, i.e. where is Italy positional in the road to reopening. But they cannot determine that from the paper as it only provide key quantities as a function of the time elapsed since the start of the vaccination. However, this time is calculated under the assumption the daily number of people being vaccinated will remain constant over time. This clearly has not been the case, with an increase in vaccine roll out in many countries similar to Italy. For example, the current vaccination rate in Italy appears to be twice that used in the model. As a result, the timeline provided by the model seems wrong, with the coverage achieved in Italy being larger than that anticipated by the model at a given time. It seems essential authors find a way to help readers understand what it means to be x months after the start of vaccination, for example by showing the associated vaccine coverage. They also need to carefully think about the relevance of a scenario with a constant vaccination rate, when this is clearly not the case. This is also the reason why the abstract which provides conditions for reopening within 9 to 15 months may seem not completely rooted in real-world.

We thank the reviewer for this comment. In the revised manuscript, we realistically implemented the vaccination campaign by feeding the model with data on the daily number of first doses administered by age group over the period analyzed (December 27, 2020 - June 30, 2021). Second doses are assumed to be administered after an average of 42 days since the first dose as was the norm for most of 2020-2021 in Italy. As shown in Figure S6A (included below for the reviewer’s convenience), the model well reproduces the observed scale-up of the daily vaccination capacity occurred in Italy.

2) Assumptions about vaccine effectiveness: Authors only account for the fact that vaccines reduce the risk of infection. There is now also evidence that vaccines provides a large protection against the risk of severe infection (higher than that obtained against the risk of infection) and that it also reduces infectivity. It seems important that authors account for the last data on vaccine effectiveness in their assessments. They should revise their model assumptions accordingly.

We thank the reviewer for this useful comment. In the revised manuscript, we consider a higher vaccine efficacy against death (as compared to infection), which was informed with data reported by the Italian integrated surveillance system [Istituto

Superiore di Sanità, bulletin of July 28]. Moreover, as suggested, we now consider a reduced infectiousness of breakthrough infections (as compared to unvaccinated infected individuals) [Harris et al., *New Eng J Med*, 2021; Lipsitch & Kahn, *Vaccine*, 2021].

In addition, we have updated the values of the age-specific vaccine efficacy against infection. To obtain these values, we weighted the efficacy of a specific vaccine type (mRNA vs. viral vectors) by the number of vaccines of that type administered to each age group in the first half of 2021 (left panel below). As shown in the figure below, the obtained values range between 70.6% and 78.8% after the first dose (average 75.5%) and between 79.4% and 88.7% after 2 doses (average: 84.9%). This figure has been added to the manuscript as well (Fig. S7).

Finally, please note that to address a comment of reviewer #1, we also changed our baseline assumption on the duration of natural (as well as vaccine-induced) immunity by assuming an average duration of 2 years, exponentially distributed. Alternative assumptions on the duration vaccine and natural immunity are explored as sensitivity analyses.

The model is capable to well reproduce the proportion of SARS-CoV-2 cases observed among individuals with complete vaccination over the whole period considered (Figure 1C in the revised manuscript, see below).

In addition, the model estimates that 97.1% (95%CI: 96.9 – 97.3%) of deaths occurring between January and June 2021 were among unvaccinated individuals, which compares well with the observed value of 95.4% (Figure 1B in the revised version, see below).

3) Description of assumptions in the main text. The description of the model and its assumption in the main text is too short; and readers need to read the Supplement to understand the work being done. It is important that key information are provided in the main text.

We apologize for the lack of detail. We have now updated and expanded the model description in the main text, including all essential information and specifically all points made by the reviewer in the following comments.

For example, the assumptions about vaccine effectiveness, which are central to the work, are only provided in the Supplement.

The age-specific vaccine efficacy against infection was re-estimated as described above. Vaccine efficacy against death was set according to estimates from the Italian integrated surveillance system. The reduction of infectiousness in breakthrough infections was set to 50% based on published estimates [Harris et al., *New Eng J Med*, 2021; Lipsitch & Kahn, *Vaccine*, 2021]. This information has been included in the main text.

Similarly, which population is targeted for vaccination? Is it really assumed that the vaccination coverage will be the same in older individuals and in young adults/teenagers/children if vaccinated?

In the revised manuscript, vaccination is administered to each age group according to actual vaccination rollout data. For prospective simulation scenarios, we assume that the same coverage will be achieved in all age groups who are below a given threshold Ω by September 7, 2021, while the coverage of remaining age groups are assumed to remain unchanged. This is because age groups with an already high coverage have remained substantially unchanged between June 30 and September 7, 2021, as shown in Figure 3B. To clarify the definition of prospective scenarios, we have added a new figure in the main text (Figure 4A). The two panels are included below for the reviewer's convenience.

Panel 4A:

Panel 3B:

What are the assumptions about R0 for the historical virus and B.1.1.7?

The value of R0 for the historical lineages was assumed to be 3.0, as estimated in the early phase of the first SARS-CoV-2 wave in several Italian regions [Cereda et al, arXiv, 2020; Riccardo et al., Eurosurv, 2020; Guzzetta et al, Em. Inf. Dis., 2021]. As for the Alpha variant (formerly known as B.1.1.7), we assumed a 50% increase in transmissibility with respect to the historical lineages [Stefanelli et al, Eurosurv., in press; Davis et al, Science, 2021; Volz et al., Nature, 2021], resulting in an R0 of 4.5. In addition, for the Delta variant we considered a further 50% increase of transmissibility with respect to that of the Alpha variant [Campbell et al, Eurosurv, 2021; Keeling, UK government, 2021; Alizon, Eurosurv, 2021], for an R0 of approximately 6.75. In sensitivity analyses, we additionally evaluated alternative values of 25% and 75% for the increase of transmissibility of the Delta variant. The considered value of R0 is now clearly stated in the main text.

When authors talk about a vaccination coverage of 75% is it across the whole population? >6 years old?

We apologize for the lack of clarity. In the revised manuscript, we only refer to age-specific vaccination coverages.

4) Authors quantify the effort required to maintain the number of cases constant with their variable “relaxation of PDRs” that is equal to 0 to do the work for the historical virus and 100% corresponds to complete relaxation. I think that for many readers it will be difficult to relate to this variable. For example, does a relaxation of 0 corresponds to the Italian lockdown of the first wave? Or less stringent measures? I wonder if it wouldn't be easier to follow if authors presented the reduction in transmission rates that is required, 0 would mean no contact at all, 100% would be all contacts restored.

We thank the reviewer for the useful suggestion, which we have adopted for the revised manuscript. We now refer to the proportion of pre-pandemic social contacts that are active (range 0-100%). The interpretation of this quantity is also more inclusive, as it accounts for changes in contacts due not only to adjustments of governmental restrictive measures (defined as physical distancing restrictions, PDRs, in the initial submission), but also to individual choices linked to the perception of risks.

5) Authors should comment in discussion about risks associated with other variants, in particular the delta variant.

The Delta variant has become dominant in Italy during the month of July 2021. For this reason, we provide indications for the expected reproduction number under different combinations on future vaccination coverage and level of social activity (expressed as the proportion of pre-pandemic contacts), by assuming as a baseline the circulation of a variant with transmissibility 50% higher than the Alpha variant, and two alternative increases in transmissibility of 25% and 75% as sensitivity analyses.

6) What are current vaccination coverages by age group in Italy? Isn't the one among older individuals already above 75%? And if so, would it be relevant to increase it?

The reviewer is correct: the coverage achieved among the elderly (80+ years) by June 30, 2021, was around 93%. As reported above, the model is now informed with actual data on the age-specific coverage of vaccination. In the revised manuscript, prospective scenarios consider a vaccination coverage equal to Ω in all age groups that have not already exceeded that coverage in September 2021, and equal to that of September 2021 otherwise.

7) Figure 4 remains difficult to follow.

The figure has been removed.

Other comments:

Abstract "A short-lived vaccine protection will require re-vaccination campaigns in absence of which a new intensification of PDRs is expected around month 14". Presumably, the month at which this occurs will be highly dependent on the duration of vaccine protection and the rhythm at which the population was vaccinated. Authors should either detail their assumptions or make a statement that remains more general.

The abstract has been completely rewritten.

Figure 1 Average relaxation of PDRs over 2 years is the average over time of the variable represented in Panel 1B, is that correct?

The figure has been removed.

Appendix line 128: "Astra Zeneca vaccine for the population under 55 years" in many countries this vaccine is no longer distributed to those aged <55 years. Revise?

See response to major point #2. We thank the reviewer for pointing this out.

It looks like, in their baseline scenario, authors only consider the impact of the vaccine on the risk of infection, not severity. There is now good evidence that SARS-CoV-2 vaccines are very effective against severe disease. It is essential that authors account for this characteristic in their main analysis.

See response to major point #2. We thank the reviewer for pointing this out.

Appendix line 192: Vaccination capacity of 4 per 1000, which corresponds to about 250,000 doses per day for a country of the size of Italy. A potential limitation with this approach is that the vaccination rate has increased dramatically in the last few months. While this rate was

elevated for many European countries some time ago, current vaccination rates can now be much higher than this.

See response to major point #1. We thank the reviewer for pointing this out.

Line 66: please give R_0 assumed for the historical lineage

This has been made explicit in the Methods.

Line 84: the number of death after 2 years without vaccination is completely dependent on the daily number of cases assumed by authors right? What was the rationale to select that number?

This is no longer the case in the revised analysis.

Figure 3B: I don't understand why, if we completely reopen at the time when the only the most fragile have been vaccinated, we have a huge drop in R , that goes below 1. Is there a problem with the way the Figure was plotted? Or is it really the case that it would be a good idea to relax everything at this stage?

This is no longer the case in the revised analysis.

In Figures, authors need to indicate what a relaxation=0 represents.

See response to major point #4. We thank the reviewer for pointing this out.

Reviewers' Comments:

Reviewer #1:

Remarks to the Author:

The authors are thanked for their extremely comprehensive response and revision of the paper. The changes to waning immunity assumptions, "zero COVID", and related changes have all enormously strengthened the paper.

Reviewer #2:

Remarks to the Author:

I'm happy with changes made in the revised manuscript and I would like to congratulate the authors for the nice work and additional analyses.

Minor comments :

1) It would be good if authors could reference few of the related modelling studies including:

- Leung et al, Effects of adjusting public health, travel and social measures during the roll-out of COVID-19 vaccination: a modelling study. Lancet Public Health 2021.

- Moore et al, Vaccination and non-pharmaceutical interventions for COVID-19: a mathematical modelling study. Lancet ID 2021.

- Tran Kiem et al, A modelling study investigating short and medium-term challenges for COVID-19 vaccination: From prioritization to the relaxation of measures. EClinicalMedicine 2021.

2) The criteria to allow relaxation is that R remains below 1. However, it is possible that even in scenarios where R is above 1, the impact on hospitals remains manageable if frail individuals are well protected by the vaccine so that hospitalizations remain limited. It would be good to comment about that point in the discussion.

3) Y-axis of Figure 2 : symbol – has been removed in "pre-pandemic". Same with age groups. Same in other figures.

REVIEWERS' COMMENTS

Reviewer #1 (Remarks to the Author):

The authors are thanked for their extremely comprehensive response and revision of the paper. The changes to waning immunity assumptions, "zero COVID", and related changes have all enormously strengthened the paper.

We thank the reviewer for giving us the chance to improve and strengthen our study. We are delighted that the reviewer is satisfied with the new version of the manuscript, and we thank the reviewer for the appreciation of our work.

Reviewer #2 (Remarks to the Author):

I'm happy with changes made in the revised manuscript and I would like to congratulate the authors for the nice work and additional analyses.

We thank the reviewer again for the useful comments provided and for the appreciation of our work.

Minor comments :

1) It would be good if authors could reference few of the related modelling studies including:
- Leung et al, Effects of adjusting public health, travel and social measures during the roll-out of COVID-19 vaccination: a modelling study. Lancet Public Health 2021.
- Moore et al, Vaccination and non-pharmaceutical interventions for COVID-19: a mathematical modelling study. Lancet ID 2021.
- Tran Kiem et al, A modelling study investigating short and medium-term challenges for COVID-19 vaccination: From prioritization to the relaxation of measures. EClinicalMedicine 2021.

The references indicated by the reviewer have now been added to the main text.

2) The criteria to allow relaxation is that R remains below 1. However, it is possible that even in scenarios where R is above 1, the impact on hospitals remains manageable if frail individuals are well protected by the vaccine so that hospitalizations remain limited. It would be good to comment about that point in the discussion.

We thank the reviewer for this comment. We have added the following sentence to the discussion:

"If the reproduction number is slightly above 1, hospitalizations could remain limited and the impact on hospitals manageable provided that frail individuals are sufficiently protected by the vaccine."

3) Y-axis of Figure 2 : symbol – has been removed in "pre-pandemic". Same with age groups. Same in other figures.

Fixed. We apologize for this visualization mistake.